# Exploring the Effects of Nitrogen and Potassium on the Aromatic Characteristics of Ginseng Roots Using Non-Targeted Metabolomics Based on GC-MS and Multivariate Analysis

**DOI:** 10.3390/foods14172981

**Published:** 2025-08-26

**Authors:** Weiyu Cao, Hai Sun, Cai Shao, Hongjie Long, Yanmei Cui, Changwei Sun, Yayu Zhang

**Affiliations:** 1Jilin Provincial Key Laboratory of Traditional Chinese Medicinal Materials Cultivation and Propagation, Institute of Special Economic Animal and Plant Sciences, Chinese Academy of Agricultural Sciences, Changchun 130112, China; 82101231147@caas.cn (W.C.); caassh@126.com (H.S.); shaocai2003@163.com (C.S.); longhongjie99@163.com (H.L.); 17860774328@163.com (Y.C.); sunchangwei@caas.cn (C.S.); 2School of Pharmacy and School of Biological Engineering, Chengdu University, Chengdu 610106, China

**Keywords:** ginseng, GC–MS, VOCs, rOAV, nitrogen–potassium interaction, flavor quality, aroma profiling

## Abstract

This study investigated individual/combined nitrogen (N) and potassium (K) deficiencies on ginseng root aroma using GC–MS metabolomics. Four treatments (normal supply, N deficiency (LN), K deficiency (LK), and dual deficiency (LNLK)) were analyzed. Deficiencies impaired growth, mineral accumulation, and induced oxidative stress, suppressing ginsenoside biosynthesis. From 1768 detected VOCs, 304 compounds (rOAV ≥ 1) significantly contributed to aroma. LN inhibited terpenoids (e.g., isoborneol) but upregulated sulfur compounds (e.g., di-2-propenyl tetrasulfide), intensifying pungency. LK enhanced sweet/woody notes (e.g., 2′-acetonaphthone) via flavonoid biosynthesis and toluene degradation. LNLK reduced esters (e.g., benzyl acetate) and terpenes, attenuating floral–balsamic nuances by coordinating aromatic degradation, glutathione metabolism, and ABC transporters. N–K nutrition dynamically shapes ginseng aroma by differentially regulating phenylpropanoid, terpenoid, and sulfur pathways, providing a foundation for precision fertilization and quality improvement.

## 1. Introduction

*Panax ginseng* C.A. Mey. (Araliaceae), historically revered as Huangjing, Dijing, Divine Grass, or the “King of Herbs,” constitutes one of the “Three Treasures of Northeast China” and is recognized as a premium medicinal and dietary resource [1]. Advances in modern scientific technologies have enabled global researchers to unveil broader pharmacological potentials, expanding ginseng applications from food/nutraceutical sectors to clinical therapeutics [2,3]. More than 300 bioactive metabolites—including polysaccharides, ginseng peptides, ginsenosides, flavonoids [4], volatile oils, organic acids, alkaloids, trace elements, and vitamins—have been identified [5]. Ginseng exerts multifaceted bioactivities, such as antioxidant [6], anti-inflammatory [7], antiallergic [8], antihypertensive [9], antiobesity [10], antidiabetic [11], antitumor [12], and cognitive enhancement effects [13].

The medicinal value of ginseng extends beyond metabolites such as polysaccharides and ginsenosides, with volatile oils constituting indispensable bioactive components. Ginseng roots are rich in volatile organic compounds (VOCs) with diverse pharmacological actions, such as anti-inflammatory, antitussive, antifatigue, hypolipidemic, ethanol detoxification, central nervous system stimulation, and antitumor activities [14,15,16,17,18]. These volatile oils underpin the characteristic aroma of ginseng, conferring distinctive flavor profiles while critically influencing pharmacological efficacy and commercial value. As a model dual-use resource (medicine–food homology), ginseng roots serve not only as essential components of traditional medicine but also as key ingredients in functional foods and premium nutraceuticals. Clinical evidence supporting lifespan-extending effects of bioactive constituents has boosted the popularity of ginseng-based functional foods [19]. Contemporary health-conscious markets have shifted demand from solely medicinal efficacy toward dual-dimensional “efficacy-flavor” criteria, with consumers increasingly valuing sensory attributes. The flavor profile of ginseng features earthy, woody, molasses-like notes accompanied by astringency, bitterness, and sweetness. Consumer acceptance is strongly influenced by sensory quality. Sweet, woody, and floral–fruity notes enhance palatability, while pungent odors may reduce commercial value. The inherent bitterness is a key factor that diminishes consumer interest and restricts food applications. Previous studies predominantly focused on pharmacological mechanisms, leaving the sensory characteristics—particularly flavor attributes—of ginseng food products substantially underexplored.

Ginseng growth and development are highly contingent upon nutrient supply, which critically governs flavor quality and overall phytochemical composition. Nutrients are essential for metabolic regulation and plant morphogenesis, with root system development directly dependent on nutrient availability [20]. Similarly, the stability and controllability of ginseng flavor attributes are strongly nutrient-dependent. Advances in metabolomics [21] have brought ginseng volatiles into the research spotlight, considerably increasing their prominence [22]. Studies confirm that terpenoids, aldehydes, esters, and other VOCs not only define the flavor profile of ginseng but also synergistically enhance its medicinal efficacy through antimicrobial, antioxidant, and neuromodulatory actions [23]. The biosynthesis of volatile compounds is modulated by environmental factors (e.g., light and temperature) and nutritional regimes. Under stress conditions—including abiotic (temperature and drought) and biotic (herbivory and pathogens) stressors—plants produce diverse VOCs as defense mechanisms against environmental challenges, facilitating ecological interactions [24,25,26]. Nitrogen (N) and potassium (K), as essential macronutrients, exert fundamental roles in plant physiological metabolism. N constitutes nucleic acids, proteins, and enzymes, directly participating in photosynthesis, respiration, and protein biosynthesis [27,28,29]. K facilitates osmoregulation, membrane potential maintenance, photosynthetic optimization, and stress resilience enhancement, critically improving water-use efficiency and stress tolerance [30,31,32]. As primary macronutrients, N and K participate in secondary metabolic networks by regulating amino acid metabolism and enzyme activities, respectively, with their impacts on ginseng growth being extensively documented [33,34,35]. However, nutrient deficiencies extend beyond physiological parameters to encompass volatile compound profiles.

Currently, the impact of N–K nutrition on the biosynthetic pathways of root aroma compounds and their key markers remains elusive, with regulatory mechanisms underlying VOC-mediated aroma metabolism in ginseng roots constituting uncharted territory. Therefore, this study employed 3-year-old ginseng roots in hydroponic experiments with individual and combined N–K deficiency (LNLK) treatments, utilizing gas chromatography–mass spectrometry (GC–MS) coupled with untargeted metabolomics and multivariate statistics to systematically decode VOC compositional shifts under N–K interactions. Key aroma markers co-modulated by N–K nutrition were identified to elucidate the regulatory mechanisms governing ginseng flavor formation. These findings establish a theoretical foundation for precision fertilization strategies in ginseng cultivation while offering novel mechanistic insights into nutrient-secondary metabolism crosstalk. By pioneering new approaches in flavor-oriented phytochemical research and value enhancement, this study ultimately facilitates targeted quality improvements in medicinal materials.

## 2. Materials and Methods

### 2.1. Plant Material and Treatments

Given the N–K interaction in ginseng plants, hydroponic experiments were conducted to investigate its effects on root VOCs. Four treatments were applied: normal N–K supply (CK), N deficiency (LN), K deficiency (LK), and LNLK. Nutrient-deficient treatments were prepared by omitting element-specific salts from full-strength Hoagland solution: CaCl_2_ substituted for Ca (NO_3_)_2_ in LN, while NaCl replaced KCl in LK. Ginseng seedlings were rinsed with deionized water and blotted dry with filter paper, then transferred to customized tanks (60 cm × 40 cm × 20 cm) containing distilled water with roots fully submerged. When plant height reached 5–6 cm, stems were clamped with cylindrical sponges (Φ 2 cm × H 2.5 cm) and inserted into predrilled holes (Φ 1.7 cm) on plastic boards (59 cm × 11 cm). Each board held 18 seedlings, with three replicate tanks per treatment. Destructive sampling for VOC analysis was performed after 40 days of cultivation [36].

### 2.2. Determination of Ginsenoside Content

#### 2.2.1. Monomeric Ginsenoside Quantification

Monomeric ginsenoside content was analyzed using methanol extraction combined with high-performance liquid chromatography.

Reference standard preparation: Nine reference standards, each accurately weighed at 10.00 mg, were volumetrically transferred to 100 mL flasks. After dissolution in methanol, stock solutions were diluted to generate gradient mixed ginsenoside standards (0.030–8.300 μg/L).

Preparation of test sample solution: A 0.1 g weight (accurate to 0.1 mg) of ginseng powder was added to a 35 mL screw-cap hydrolysis tube. A 25.00 mL volume of methanol–water (9 + 1) extraction agent was then added, the screw cap sealed, and the mixture soaked overnight. It was then sonicated at 60 °C for 1 h. After mixing, the sample solution was filtered through a 0.2 μm nylon 66 ultrafiltration membrane for injection. For high-content components, the sample solution was diluted with methanol 10-fold or 100-fold, then filtered through a 0.2 μm nylon 66 ultrafiltration membrane before injection.

Chromatographic analysis was conducted using a BEH C18 column (Waters, Milford, CT, USA) (1.7 μm, 2.1 × 100 mm) at a temperature of 45 °C. The mobile phase consisted of acetonitrile (A) and water (B), with a flow rate of 0.5 mL/min and an injection volume of 3 μL. The gradient program was as follows: 0–4.5 min at 19% A, 4.5–8.8 min transitioning from 19% to 28% A, and 8.8–17 min increasing from 28% to 37% A.

The mass spectrometric analysis was performed under the following conditions: the ionization mode was set to ESI-negative, with a capillary voltage of 2.80 kV. The desolvation gas used was N_2_ with a purity of ≥95% at a flow rate of 1000 L/h, and the source temperature was maintained at 450 °C.

#### 2.2.2. Total Ginsenoside Determination

The dried ginseng root samples were ground and pulverized, then sieved through a 60-mesh screen. A 1.0 g weight of ginseng root powder was accurately weighed, extracted with ether using the Soxhlet method for 1 h, and subjected to methanol reflux extraction until saponins were completely extracted (verified by TLC). The extract was evaporated to dryness, dissolved in water, and extracted four times with water-saturated n-butanol. The n-butanol phases were combined, and the residue diluted with methanol to 10 mL. A 20 μL volume of the test solution was evaporated to dryness, and then 0.5 mL of 8% vanillin ethanol solution and 5 mL of 72% sulfuric acid were added. The color was developed at 60 °C in a water bath for 10 min. The reaction was stopped with ice water. Measurements were performed at a wavelength of 544 nm using a UV spectrophotometer (TU-1950) (GB/T 18765-2015) [37].

### 2.3. Determination of Mineral Elements

After drying the ginseng roots, they were crushed and passed through a 100-mesh sieve. A 0.1000 g weight of each ginseng root was weighed and placed in a 100 mL conical flask, and 10 mL of concentrated nitric acid was added. A small funnel was placed on top of the conical flask, which was then placed on a heating plate and heated at 80 °C for 30 min. The temperature was gradually increased to 160 °C, and when the brownish-red gas at the mouth of the flask disappeared, 2.5 mL of perchloric acid was added. The temperature was then raised to 180 °C, and the flask was boiled until the liquid inside became transparent. After cooling, the solution was made up to 50 mL with distilled water in a 50 mL volumetric flask and then filtered into a 50 mL conical flask. The digested samples were then analyzed for potassium (flame photometry) and phosphorus (sodium hydroxide fusion–molybdenum antimony colorimetry). Organic matter (OM) and total nitrogen (TN) were determined using an elemental analyzer Vario ELIII (EA300, Eurovector, Reggio Emilia, Italy).

### 2.4. Measurement of Antioxidant Enzyme Activities

Antioxidant assays: Catalase (CAT), superoxide dismutase (SOD), malondialdehyde (MDA), hydrogen peroxide (H_2_O_2_), DPPH radical scavenging capacity, and superoxide anion (O_2_^−^) levels were measured using commercial kits from Beijing Box Biotechnology Co., Ltd. (China). Rubisco activity was quantified using ELISA kits (EnzymeLink Biotechnology Co., Ltd., Shanghai, China).

### 2.5. Sample Preparation for VOC Analysis

#### 2.5.1. Headspace Solid-Phase Microextraction

Materials were harvested, weighed, immediately frozen in liquid nitrogen, and stored at −80 °C until needed. The samples were ground to a fine powder in liquid nitrogen. Briefly, 500 mg of the powder was transferred immediately to a 20 mL headspace vial (Agilent, Palo Alto, CA, USA) containing a NaCl-saturated solution to inhibit enzymatic reactions. The vials were sealed using crimp-top caps with TFE-silicone headspace septa (Agilent). For SPME analysis, each vial was incubated at 60 °C for 5 min, followed by exposure of the sample headspace to a 120 µm DVB/CWR/PDMS fiber (Agilent) for 15 min at 60 °C.

#### 2.5.2. GC–MS Conditions

After sampling, VOC desorption from the fiber coating was performed in the injection port of the GC apparatus (Model 8890; Agilent, CA, USA) at 250 °C for 5 min in the splitless mode. VOC identification and quantification were conducted using an Agilent Model 8890 GC coupled with a 7000D mass spectrometer equipped with a 30 m × 0.25 mm × 0.25 μm DB-5MS capillary column (5% phenyl-polymethylsiloxane). Helium served as the carrier gas at a linear velocity of 1.2 mL/min, with the injector temperature being maintained at 250 °C, splitless injection, and a solvent delay of 3.5 min. The oven temperature was programmed as follows: starting at 40 °C for 3.5 min, then increasing at 10 °C/min to 100 °C, at 7 °C/min to 180 °C, and at 25 °C/min to 280 °C, and holding for 5 min. Mass spectra were recorded in the electron impact ionization mode at 70 eV. The quadrupole mass detector, ion source, and transfer line temperatures were 150 °C, 230 °C, and 280 °C, respectively. The MS was operated in the ion monitoring mode for analyte identification and quantification.

### 2.6. Identification and Quantification of VOCs

VOC qualitative analysis was performed using Agilent’s MassHunter 12.0 software. The deconvolution parameter peak width was set to 20, and the resolution, sensitivity, and chromatographic peak shape requirements were all set to medium. The minimum matching factor was set to 70. Substance identification was performed by comparing the obtained mass spectrometry data with the mass spectra of standard substances provided by the NIST (2020) library. The retention index (RI) of each volatile compound was then calculated using an n-alkane mixture (C7–C40) as the standard. Under the same chromatographic conditions, gas chromatography–mass spectrometry (GC-MS) analysis was performed, and the compounds were calculated using the following equation, which was compared with the RI values reported in the literature. Additionally, further qualitative confirmation was conducted based on reports in the literature. Through the above methods, a specialized volatile broad-target database was established. During the detection process, substances with retention times (RTs) and characteristic fragments that met the parameter settings were considered target substances (GB 23200.8-2016). The internal standard used was 20 μL (10 μg/mL) of 3-Hexanone-2,2,4,4-d4 (3-hexanone-2,2,4,4-d4). Based on the content of 3-Hexanone-2,2,4,4-d4 (3-hexanone isotope internal standard), the chromatographic peak area of the analyte was compared with that of the internal standard, and the relative content was calculated using the peak area normalization method.

### 2.7. Relative Odor Activity Value Calculation

Relative odor activity value (rOAV) is a method based on the sensory threshold of compounds used to identify key flavor compounds in food and determine their contribution to the overall aroma characteristics of a sample. In recent years, scholars have increasingly applied rOAV to identify key flavor compounds in various foods. Generally, an rOAV of ≥1 indicates that a compound directly contributes to the flavor of the sample [38,39,40]. rOAV is expressed as follows:$$rOAV_{i}=\frac{C_{i}}{T_{i}}$$
where rOAV_i_ represents the relative odor activity value of compound i, C_i_ denotes its relative content (ug/mL), and T_i_ represents its threshold (ug/mL).

## 3. Statistical and Multivariate Data Analyses

To ensure the reliability of the experimental results, three experiments were conducted for each group of samples. The GC–MS results are presented as the mean ± standard deviation; data without standard deviation indicates that the test value is missing. Missing values were filled in using 1/5 of the minimum value measured in the four treatment groups to facilitate subsequent data analysis and graphing without affecting trend changes. Simca software was used to perform principal component analysis (PCA), orthogonal partial least squares-discriminant analysis (OPLS-DA), and variable importance in projection (VIP) value analysis. The heatmap was generated using z-score values from the metabolomics dataset and plotted on a free online platform (https://cloud.metware.cn, accessed on 18 May 2025). Kyoto Encyclopedia of Genes and Genomes (KEGG) annotations and metabolic pathway analyses were conducted for differential metabolites. KEGG serves as a primary public pathway-related database, encompassing genes and metabolites. Metabolites were mapped to KEGG metabolic pathways for pathway and enrichment analysis. Pathway enrichment analysis identified significantly enriched metabolic or signal transduction pathways in differential metabolites compared to the background. Identified metabolites were annotated using the KEGG compound database (http://www.kegg.jp/kegg/compound, accessed on 7 May 2025/) and subsequently mapped to the KEGG pathway database (http://www.kegg.jp/kegg/pathway.html, accessed on 7 May 2025). The calculation formula is expressed as follows:P=1−∑i=0m−1MiN−Mn−iNn
where N represents the total number of metabolites with KEGG annotation, *n* denotes the number of differential metabolites within N, M represents the total number of metabolites annotated to specific pathways, and m denotes the number of differential metabolites within M. The calculated *p*-value was adjusted using FDR correction, with a threshold of FDR ≤ 0.05. Pathways meeting this condition were defined as significantly enriched pathways in differential metabolites.

## 4. Results

### 4.1. N and K Deficiency Effects on N, Phosphorus, K, and OM Contents in Ginseng Roots

Under sufficient nutrient supply, the roots of *P. ginseng* exhibited relatively high levels of total carbon, TN, total phosphorus (TP), and total potassium (TK), highlighting the importance of adequate N and K supply for ginseng growth and mineral accumulation. As shown in Figure 1, the LN treatment caused a significant decrease in TN content, while the LK treatment significantly reduced TK content and led to a decline in TP content. This suggests that N and K play crucial roles in phosphorus absorption and utilization in ginseng. Under the combined LNLK treatment, the levels of N, phosphorus, K, and OM in ginseng roots were reduced to their lowest values.

### 4.2. Effects of N and K Deficiency on Ginsenoside Content in Ginseng Roots

Figure 2 shows the TG content and the levels of various individual ginsenosides (Rb1, Rb2, Rb3, Rc, Rd, Re, Rg1, and Rg2) in the roots of *P. ginseng* under four different treatments. Under normal CK conditions, the TG and individual ginsenoside contents in the ginseng roots reached their highest levels, indicating that adequate nutrient supply promotes comprehensive synthesis and accumulation of ginsenosides. In the LN group, TG content and most individual ginsenosides (Rb1, Rb2, Rb3, Rc, Rd, and Rg1) significantly decreased, suggesting that LN influences enzyme activities and metabolic pathways involved in ginseng saponin synthesis, thereby inhibiting ginsenoside production. Additionally, the content of Re saponin decreased under N-deficient treatment; however, its decline was relatively smaller, which may indicate that the synthesis of Re saponin is less dependent on N. Under LK treatment, the total saponin and individual ginsenoside contents significantly decreased compared to those in the normal nutrient supply group, although the decline was less pronounced than that under N deficiency. This may be because K primarily regulates osmotic pressure and ion balance in plant cells, with a relatively smaller direct impact on saponin synthesis. However, LK still negatively affects ginseng growth and metabolism, indirectly influencing saponin production. Under the LNLK treatment, the contents of total saponins and individual ginsenosides in the roots of ginseng were reduced to the lowest levels. This result further confirms the synergistic effect of N and K on ginsenoside synthesis. In the absence of these two key nutrients, the growth and metabolism of ginseng are severely inhibited, leading to a significant decline in saponin synthesis capacity.

### 4.3. Effects of N and K Deficiency on the Activity of Antioxidant Enzymes in Ginseng

Under normal CK conditions (Figure 3), ginseng exhibited enhanced antioxidant capacity with attenuated oxidative stress levels. During N/K deficiencies, elevated activities of SOD, CAT, and MDA, along with increased DPPH scavenging rates, were observed owing to their critical roles in eliminating reactive oxygen species (ROS) generated by abiotic stress, indicating that N–K nutrition is essential for sustaining antioxidant competence and mitigating oxidative stress. Nutrient deprivation further intensified oxidative stress by augmenting specific ROS types, including superoxide anions (O_2_^−^) and H_2_O_2_. Under LN, LK, and LNLK treatments, enzyme activity alterations were significantly intensified, leading to a progressive decline in antioxidant capacity and heightened oxidative stress.

### 4.4. Volatile Compound Analysis

#### 4.4.1. Volatile Compounds in Ginseng Roots Under Different Treatments

To better understand the metabolic differences among the roots of ginseng under different treatments, GC–MS metabolite analysis was conducted on the roots of ginseng under four treatments using optimized experimental parameters, and the metabolites in the samples were identified. The total ion chromatograms of volatile metabolites of ginseng under the four treatments analyzed by SPME-GC-MS are shown in Figure 4. QC was prepared by mixing samples and was used to analyze the repeatability of the samples under the same treatment method. During instrument analysis, a quality control sample was inserted every 10 analyzed samples to monitor the repeatability of the analysis process. The superimposed total ion chromatogram (TIC) of the QC sample mass spectrometry detection shows that the curves of the total ion flow of the metabolites overlap well; that is, the retention time and peak intensity are consistent, indicating that the signal stability of the mass spectrometry for the same sample at different times is good. In the sample CV distribution diagram, the two reference lines perpendicular to the X-axis correspond to CV values of 0.3 and 0.5, and the two reference lines parallel to the X-axis correspond to the proportion of substances accounting for 75% and 85% of the total substances. CV value, or coefficient of variation, is the ratio of the standard deviation of the original data to the average of the original data, which can reflect the degree of data dispersion. The empirical cumulative distribution function (ECDF) can be used to analyze the frequency of occurrence of substances with CV values less than the reference value. The higher the proportion of substances with lower CV values in the QC sample, the more stable the experimental data, and in this experiment, the proportion of substances with CV values less than 0.3 in the QC sample is higher than 75%, indicating that the experimental data is very stable and the experimental repeatability is good.

In total, 1768 metabolites were detected and classified into 15 categories (Figure 5A), including 398 terpenoids, 339 esters, 182 ketones, 168 heterocyclic compounds, 146 alcohols, 102 aldehydes, 91 carboxylic acids, 81 phenols, 67 hydrocarbons, 56 amines, 55 aromatic hydrocarbons, 45 ethers, 24 N-containing compounds, 8 halogenated hydrocarbons, and 6 sulfur-containing compounds (Appendix A). While the types of metabolites were similar across all treatment groups, their relative abundances differed significantly. Collectively, these results indicate distinct metabolic profiles among the treatments. To eliminate scaling effects on pattern recognition, the peak area of each metabolite was log10-transformed and subjected to hierarchical clustering analysis (Figure 5B). The resulting heatmap revealed four distinct clusters, demonstrating that ginseng roots under the four treatments exhibited divergent metabolite profiles. These differential results suggest that the treatments strongly influenced the metabolic composition of ginseng roots.

#### 4.4.2. Multivariate Statistical Analysis of VOC Metabolic Profiles

PCA reduces the dimensionality of metabolomics data by extracting principal components (PCs) to reveal the internal structure of multivariate datasets. PCA score plots thus enable the visualization of intergroup differences. Unsupervised PCA was performed to assess variations between the treatment groups, within the groups, and among the individual samples (Figure 6A). The first two PCs accounted for 36.15% (PC1) and 25.35% (PC2) of the total variance, respectively, with a cumulative contribution rate of 61.50%. Distinct separation among the four treatment groups was observed in the 2D score plot, without detecting outliers. The samples from the same treatment clustered tightly, indicating significant metabolic divergence between the treatments. Quality control samples, prepared as pooled mixtures of ginseng root extracts, were projected within a confined region of the PCA space, with partial overlap indicating metabolic similarity. This clustering consistency validates analytical stability and reproducibility, supporting further downstream analyses. Correlation analysis (Figure 6B) revealed strong intra-group reproducibility, with correlation coefficients (r) of >0.8 between the biological replicates. These results indicate that the metabolic differences induced by the different treatments are highly reliable.

### 4.5. Calculation of rOAVs of Aroma Compounds

The rOAV is a metric for identifying key flavor compounds by incorporating odor thresholds, elucidating individual contributions to the overall aroma profiles. The significance of volatile compounds depends on two factors: concentration and odor threshold. Compounds with rOAV of ≥1 are generally considered direct contributors to flavor perception [41,42]. Based on the qualitative and quantitative results obtained from GC–MS, the threshold values of the corresponding aroma compounds in water were identified through literature searches [1,2,3,4]. Compounds without reported threshold values or those below the quantitative detection limit were excluded, and their rOAVs were calculated. Among 1768 metabolites, 304 aroma compounds exhibited an rOAV of ≥1 across treatments, including 54 terpenoids, 55 esters, 30 ketones, 36 heterocyclic compounds, 31 alcohols, 44 aldehydes, 1 carboxylic acid, 23 phenols, 2 hydrocarbons, 3 amines, 15 aromatic hydrocarbons, 4 ethers, 3 N-containing compounds, 1 halogenated hydrocarbon, and 2 sulfur-containing compounds. These constitute the primary contributors to the characteristic aroma of ginseng. Figure 7 (rOAV scatter plot) shows a clustered distribution of high-rOAV compounds in the control group, suggesting enhanced contributions of specific aroma-active substances under nutrient sufficiency.

### 4.6. Key Aroma Metabolite Screening

To identify characteristic volatile compounds associated with nutrient-induced metabolic shifts in ginseng roots, a quantitative correlation model linking volatile profiles and sensory attributes was developed using PCA integrated with OPLS-DA. OPLS-DA, a supervised multivariate algorithm derived from PLS-DA, incorporates orthogonal signal correction to partition X-matrix information into Y-correlated and Y-uncorrelated components. This enhances model specificity by isolating biologically relevant variations in the first predictive component, thereby optimizing differential metabolite screening [43,44]. The OPLS-DA score plot (Figure 8A) showed clear segregation among the four treatment groups, with minimal intra-group variation and complete intergroup separation, consistent with PCA results. Permutation testing (n = 200) confirmed model robustness, as the original model’s R^2^ (0.92) and Q^2^ (0.85) exceeded permuted Y-matrix values (R^2^ = 0.21, Q^2^ = −0.38), validating its predictive reliability. Variables with VIP of ≥1 were selected as differential VOCs. The S-plot is a critical output of the OPLS model that visualizes the extent to which the variables contribute to the model (Figure 8B). Figure 8C (VIP-ranked differential metabolites) shows that the top 20 metabolites with the highest VIP values were predominantly distributed across aromatic hydrocarbons, esters, terpenoids, phenols, and alcohols.

The expression levels of naphthalene, 2-methyl-, and naphthalene, 1-methyl- were significantly upregulated in the LN and LNLK treatment groups, which might be associated with the activation of phenylpropanoid metabolism or the accumulation of microbial degradation products [45,46]. Biphenyl was simultaneously upregulated in the LN and LNLK treatment groups, suggesting that the synthesis of polycyclic aromatic hydrocarbons was enhanced or involved in antioxidant defense under low N conditions. Acetic acid, 4-methylphenyl ester, and m-cresyl acetate were significantly downregulated in the LK treatment group, indicating that low K inhibited the esterification branch of phenylpropanoid metabolism, possibly reducing the release of floral volatile compounds. Estragole was upregulated in the LNLK treatment group. As an ether product of phenylpropanoid metabolism, estragole might respond to combined stress by activating the benzoate ester synthesis pathway. Benzenepropanoic acid methyl ester was significantly downregulated in the LK and LNLK treatment groups, possibly indicating that low K inhibited the esterification process, affecting the release of aroma components. cis-1,2-Limonene oxide was significantly downregulated in the LN treatment group, suggesting that low N inhibited the activity of terpene synthase (TPS), thereby reducing monoterpene oxidation modification, which might weaken the citrus-like aroma. Tropinone was downregulated in the LNLK treatment group. As a precursor of tropane alkaloids, the reduction of tropinone may disrupt the secondary metabolism balance through N metabolism disorder. 2-(Methylthio)phenol was significantly accumulated in the LNLK treatment group; its sulfur-containing structure might respond to oxidative stress through the glucosinolate metabolism pathway, exhibiting antibacterial and stress-resistant functions. Benzenepropanal was downregulated in the LK treatment group, indicating that low K inhibited the aldehyde branch of phenylpropanoid metabolism, affecting the woody and sweet aroma characteristics. The fluctuations of esters (such as methyl phenylpropanoate) and terpenoids (such as limonene oxide) directly affected the sweet and citrus aromas of ginseng, indicating that the N–K ratio could optimize flavor by targeting the regulation of esterification enzymes or oxidase activities. The top 20 differential metabolites with VIP values revealed the complex reconfiguration of the secondary metabolism network in ginseng roots under N and K stress. The primary features included the redirection of phenylpropanoid metabolism branches (inhibition of esters and activation of ethers), enhanced detoxification of naphthalene derivatives, and regulation of volatile aldehydes mediated by lipoxygenase, demonstrating that the interaction between N and K regulates the secondary metabolism network through multiple pathways in a coordinated manner.

Differentially expressed metabolites were screened based on VIP of ≥1 and *t*-test *p* of <0.05, identifying a total of 669 metabolites across the four groups. We mapped the 669 intergroup differentially expressed metabolites to the KEGG database and first examined the pathway information (Figure 9A). Our results indicate that most metabolites are mapped to the “metabolism” pathway of secondary metabolites, aligning with our expectations. A small number of metabolites were classified under “environmental information processing” and “cellular processes.” KEGG results indicated that in the “metabolism” pathways, “metabolic pathways,” “microbial metabolism in different environments,” “biosynthesis of secondary metabolites,” and “degradation of aromatic compounds” played crucial roles. In the “environmental information processing” pathways, the “two-component system” and “ABC transporters” pathways played essential biological roles. In the “cellular processes” pathways, the “endocytosis” pathway played a significant role, suggesting its involvement in N- and K-induced volatile changes in ginseng roots. The KEGG differential enrichment bubble plot (Figure 9B) highlights “Degradation of aromatic compounds,” “Metabolic pathways,” “Glutathione metabolism,” and “ABC transporters” as key pathways with crucial biological functions in the LN, LK, and LNLK treatment groups.

### 4.7. Response of Ginseng Roots to VOCs Under N and K Deficiency

#### 4.7.1. Statistical Analysis of Intergroup Differences in VOCs

The OPLS-DA model was further applied to screen differential VOCs (Figure 10A), with volcano plots illustrating volatile metabolite expression disparities between pairwise treatment groups (Figure 10B). The number of differential metabolites between LK and CK, LN and CK, and LNLK and CK is shown in Figure 10C. A total of 244 differential metabolites (52 upregulated and 192 downregulated) were identified between LK and CK. These metabolites may represent volatile compounds that ginseng roots produce in response to K deficiency. Between LN and CK, 340 differential metabolites (99 upregulated and 241 downregulated) were detected. These metabolites are likely volatile compounds linked to the response of ginseng root to N deficiency. In the LNLK versus CK comparison, 288 differential metabolites were identified, with 60 upregulated and 228 downregulated. These differential metabolites are volatile compounds that are significantly responsive in ginseng roots under K-deficient, N-deficient, and N–K-deficient conditions.

#### 4.7.2. Response of Ginseng Roots to VOCs Under N Deficiency

Differential metabolites were screened by selecting metabolites with VIP of ≥1 and *p* < 0.05 in the OPLS-DA model, and potential N-responsive biomarkers were further identified based on fold-change (FC) values of ≥2 or ≤0.5. As shown in the heatmap of differential metabolites between the LN and CK treatment groups (Figure 11A), these metabolites effectively distinguished N-deficient from N-sufficient samples. KEGG pathway analysis (Figure 11B) revealed that most differential metabolites were mapped to the “Metabolism” pathway of secondary metabolites, while a smaller subset was classified under “Environmental Information Processing” and “Cellular Processes”. Among the top five enriched pathways, two (ko00622 and ko01220) [47] contained more than five differential metabolites, corresponding to xylene and aromatic compound degradation, respectively. Under low N conditions, plants activate aromatic compound degradation pathways. The carbon skeletons released from this degradation are integrated into central metabolism, alleviating N-deficiency-induced constraints on amino acid synthesis. These carbon skeletons may enter the gluconeogenesis pathway through the glyoxylate cycle, maintaining homeostasis between carbon and N metabolism. By degrading secondary metabolites with low N content, limited N resources are preferentially allocated to essential metabolic pathways. The primary function of the xylene degradation pathway is the stepwise oxidation of aromatic hydrocarbons into key intermediates, such as acetyl-CoA and succinate [48], which are ultimately incorporated into the tricarboxylic acid (TCA) cycle. Under N limitation, carbon skeletons from aromatic hydrocarbon degradation generate glucose and ATP through gluconeogenesis or the TCA cycle, compensating for impaired amino acid synthesis. Aromatic compounds may accumulate under stress owing to membrane lipid peroxidation or secondary metabolic dysregulation. The activation of their degradation pathways (e.g., 4-methyl-1,2-benzenediol, 3-methyl-1,2-benzenediol, o-xylene, 1,3-dimethylbenzene, and p-xylene) reflects the active detoxification strategy of plants to mitigate cellular damage. While the degradation of pungent aromatic hydrocarbons reduces malodor accumulation, aromatic compound degradation may concurrently diminish phenylpropanoid precursors, limiting the synthesis of esters (e.g., butoxybenzene) and terpenoids (e.g., isoborneol) [49], thereby compromising desirable aromatic traits in ginseng. Carbon resource recycling and detoxification through aromatic compound degradation constitute a primary strategy for ginseng to cope with N stress. This confirms that under adverse conditions, ginseng employs metabolic responses to prevent, reduce, or repair stress-induced damage, maintaining normal physiological activities [50].

Figure 11C shows the odor categories and potential sensory contributions of differential metabolites in ginseng roots under LN and CK treatments. This demonstrates that N deficiency primarily affects the contribution levels of three key flavor attributes: “sweet,” “fruity,” and “green.” Further classification of differential metabolites by sensory attributes (Appendix A) revealed that sweet-type (41 compounds), fruity-type (35 compounds), and green-type (30 compounds) metabolites dominated the differential pool, accounting for 21.4%, 18.3%, and 15.7%, respectively. This indicates significant divergence in these odor characteristics between the two groups. Floral-type (21 compounds) and woody-type (19 compounds) metabolites exhibited high abundance, suggesting their potential influence on the overall flavor profile of ginseng roots through floral–woody synergy.

Figure 11D and Table 1 show the top 20 differential metabolites ranked by FC between the LN and CK treatment groups. Among these 20 metabolites were five ketones, four heterocyclic compounds, three terpenoids, one alcohol, one ester, one aromatic hydrocarbon, one sulfur-containing compound, one N-containing compound, one phenolic compound, one aldehyde, and one ether. These metabolites may serve as signature biomarkers for ginseng roots under N deprivation.

#### 4.7.3. Response of Ginseng Roots to VOCs Under LK

Differential metabolites were similarly screened by selecting compounds with VIP of ≥1 and *p* < 0.05 in the OPLS-DA model, with potential K-responsive biomarkers identified based on FC values of ≥2 or ≤0.5. The heatmap of differential metabolites between the LK (low K) and CK treatment groups (Figure 12A) demonstrated clear separation between K-deficient and K-sufficient samples. KEGG pathway analysis (Figure 12B) revealed that most differential metabolites were mapped to the “Metabolism” category of secondary metabolites, while a smaller subset was associated with “Environmental Information Processing.” Although no pathway showed significant enrichment for LK-specific metabolites, the top five enriched KEGG pathways (>5 differential metabolites) in LK versus CK comparisons were toluene degradation (ko00623), flavonoid biosynthesis (ko00946), naphthalene degradation (ko00626), glycerophospholipid metabolism (ko00564), and pyrimidine metabolism (ko00240). These pathways reveal adaptive regulatory strategies in ginseng roots under LK, involving aromatic hydrocarbon detoxification, antioxidant defense, membrane remodeling, and metabolic trade-offs. K deprivation induced ROS accumulation in roots, where aromatic hydrocarbon degradation and flavonoid biosynthesis pathways alleviated stress by regulating K uptake/partitioning and detoxification. Notably, both processes share dependence on phenylpropanoid precursors. Under K limitation, suppressed phenylalanine ammonia lyase activity likely reduced lignin synthesis, redirecting carbon flux toward defense pathways such as toxin degradation or flavonoid production. This is corroborated by significant downregulation of methyl benzenepropanoate (a key phenylpropanoid ester) among the top 20 differential metabolites, supporting inhibition of phenylpropanoid esterification under low K. Comparative analysis with N stress responses revealed that N and K deficiencies activate aromatic compound degradation (e.g., ko00623 and ko00626), suggesting carbon skeleton recycling as a universal nutrient limitation strategy. However, N stress primarily focuses on N conservation, suppressing secondary metabolism (e.g., terpenoids) while enhancing sulfur metabolism. In contrast, K stress prioritizes membrane stabilization and antioxidant responses (flavonoids and phospholipid metabolism), highlighting the distinct roles of K in osmotic regulation and enzyme activation.

Figure 12C shows the quantitative distribution of odor categories and potential sensory contributions among differential metabolites in ginseng roots under LK and CK treatment conditions. The flavor profile revealed a primary sensory pattern dominated by sweet-type (34 compounds), fruity-type (27 compounds), and herbal-type (19 compounds) metabolites, accounting for 23.0%, 18.2%, and 12.8%, respectively. Classification by sensory attributes (Appendix A) revealed that compared with the LN treatment group, the LK treatment group exhibited significantly enhanced herbal characteristics with a co-distribution of woody-type (16 compounds) and balsamic-type (14 compounds) metabolites. This suggests a woody–balsamic complex as a signature flavor trait of the LK treatment group. Key metabolites demonstrated pleiotropic regulation properties: 1-cyclohexene-1-carboxaldehyde, 4-(1-methylethenyl)- (spanning sweet, herbal, and minty categories), and pentyl propanoate (contributing to fruity and tropical notes). These compounds, widely distributed in terpenoid and ester metabolic pathways, indicate their hub-like function in the flavor network. Among floral-type metabolites (15 compounds), α-(trichloromethyl)benzyl acetate (an ester) shared 42.9% co-occurrence with balsamic-type metabolites, implying that phenylpropanoid biosynthesis may drive floral–balsamic synergy. Low-abundance categories (e.g., aldehydic and acetone) were predominantly governed by single metabolites.

Figure 12D and Table 2 show the top 20 differential metabolites ranked by FC. The composition included six ketones, six phenolic compounds, two esters, two N-containing compounds, two alcohols, one carboxylic acid, and one terpenoid. These metabolites serve as signature biomarkers for ginseng roots under K deprivation.

#### 4.7.4. Response of Volatile Metabolites in Ginseng Roots Under Simultaneous LNLK

Differential metabolites were screened through OPLS-DA with VIP of ≥1 and *p* of <0.05, followed by identification of potential biomarkers for dual N–K deficiency using FC criteria (FC ≥ 2 or ≤0.5). The heatmap (Figure 13A) showed distinct segregation between the LNLK and CK treatment groups based on differential metabolite profiles. KEGG analysis (Figure 13B) predominantly mapped these metabolites to “Metabolism” pathways (secondary metabolites), with minor associations with “Environmental Information Processing” and “Cellular Processes.” Although no pathway showed significant enrichment (FDR > 0.05), the top five KEGG pathways ranked by *p*-value were ko00626 (naphthalene degradation), ko00623 (toluene degradation), ko00480 (glutathione metabolism), ko02010 (ABC transporters), and ko00410 (beta-alanine metabolism). These pathways represent integrated adaptive mechanisms under dual nutrient limitation. Aromatic hydrocarbon degradation (ko00623/ko00626) converts naphthalene and toluene into acetyl-CoA and succinate through hydroxylation and ring cleavage, feeding carbon skeletons into the TCA cycle to mitigate C:N imbalance, recycle carbon for energy and amino acid synthesis, and reduce membrane lipid peroxidation and cytotoxicity. Glutathione metabolism (ko00480) facilitates ROS scavenging [51], heavy metal chelation, and sulfur homeostasis to maintain membrane integrity. ABC transporters (ko02010) regulate transmembrane transport of ions, secondary metabolites, and xenobiotics, potentially alleviating LK-induced ion dysregulation. Comparative analysis revealed significant enrichment of aromatic degradation under single (LN/LK) and dual (LNLK) deficiencies, confirming carbon recycling as a universal stress response. By contrast, glutathione metabolism reflected additive oxidative damage under co-stress conditions. Notably, ABC transporters and β-alanine metabolism were exclusively enriched in the LNLK treatment groups, indicating dual deficiency demands refined metabolite partitioning and energy deficit compensation—mechanisms absent in single stresses, demonstrating more pronounced damage under LNLK treatment.

Integrated analysis of flavor characteristics in differential metabolites between the LNLK and CK treatment groups (Figure 13C) revealed a predominance of sweet (43 metabolites), fruity (34 metabolites), and green (30 metabolites) compounds in ginseng roots under LNLK. These accounted for 22.7%, 17.9%, and 15.8% of total differential metabolites, respectively—significantly higher than in single deficiency groups (LN/LK). This suggests a systematic enhancement of sweet metabolite pathways as a defining feature of LNLK. Further classification by sensory attributes (Appendix A) revealed a significant positive correlation between woody (18 metabolites) and balsamic (17 metabolites) categories. Shared metabolites (e.g., acetic acid, 1,7,7-trimethyl-bicyclo [2.2.1]hept-2-yl ester, 3-buten-2-one, and 4-phenyl-) were co-regulated through phenylpropanoid biosynthesis, suggesting woody–balsamic flavor integration as a signature regulatory module in LNLK. Key multifunctional metabolites—1-cyclohexene-1-carboxaldehyde, 4-(1-methylethenyl)- (spanning sweet/herbal/minty categories), and 2,4,6-octatriene, 2,6-dimethyl-, (E,E)- (sweet-terpenoid)—showed frequent enrichment in monoterpene/sesquiterpene synthesis pathways, indicating their role in modulating multidimensional VOC networks. Among floral compounds (27 metabolites), benzenemethanol, α-(trichloromethyl)-acetate shared 40.7% identity with balsamic category metabolites, potentially driving floral–balsamic synergy through benzoate metabolism. Low-abundance categories—aldehydic (two metabolites) and acetone (one metabolite)—were dominated by singular compounds (e.g., heptanal and cyclohexanone), likely associated with stress-induced shifts in lipoxygenase activity.

Figure 13D and Table 3 show the top 20 differential metabolites ranked by FC in K-deficient versus control groups, comprising four ketones, two terpenes, six esters, two phenols, one N-containing compound, two amines, one ether, and two aldehydes, which collectively represent signature volatile biomarkers for LNLK in ginseng roots.

## 5. Discussion

### 5.1. Physiological Metabolic Disorders Drive the Reprogramming of the Aroma Pathway

Mineral element imbalance restricts the supply of metabolic precursors. N deficiency significantly reduces TN content in ginseng roots, directly inhibiting N-dependent enzyme activities (e.g., nitrate reductase and glutamine synthetase) and depleting amino acid pools. This not only impairs protein synthesis but also diminishes key precursors (e.g., phenylalanine and leucine) essential for volatile aldehyde and ester biosynthesis. Similarly, K deficiency significantly lowers TK and TP levels, disrupting ion homeostasis and energy metabolism by impairing ATP synthesis. Consequently, energy-intensive pathways, such as TPS activity, are suppressed—consistent with the downregulation of monoterpene isoborneol in the LK treatment group. Dual N–K deficiency causes the synchronous depletion of mineral elements (TN, TK, and TP) and OM, triggering global carbon–N metabolic dysregulation that diverts carbon flux toward survival-essential pathways (e.g., aromatic hydrocarbon degradation), representing an adaptive stress response [52]. Under mineral imbalance, ginsenoside synthesis—a key secondary metabolite co-regulated by N and K [53]—is suppressed [54]. TG content significantly declined in the LN, LK, and LNLK treatment groups, with monomeric ginsenosides (Rb1 and Rg1) showing synchronized reduction, demonstrating stronger inhibition under dual deficiency than single stresses. This indicates that plants prioritize reallocating limited carbon/N resources from ginsenoside biosynthesis to stress-responsive volatile metabolism (e.g., upregulation of sweet compound 2-naphthyl methyl ketone in the LK treatment group; activation of sulfur-containing volatiles in the LNLK treatment group), embodying a critical “ginsenoside–VOCs trade-off strategy” for resource optimization. Oxidative stress directly triggers defensive aroma responses. Under sufficient CK conditions, ginseng maintains robust antioxidant capacity with minimal oxidative stress. Alternatively, deficiency conditions lead to increased superoxide anion (O_2_^−^) accumulation and a significant elevation of MDA levels in the LN, LK, and LNLK treatment groups, indicating membrane lipid peroxidation damage. Antioxidant enzymes (SOD and CAT) exhibited increased activity in single deficiency groups, while redox imbalance was exacerbated under dual deficiency. When stress exceeds the defense threshold of the plant, radical accumulation intensifies, where ROS (e.g., O_2_^−^ and H_2_O_2_) induce oxidative stress [55,56]. Notably, oxidative stress couples with sulfur metabolism activation, as evidenced by the accumulation of sulfur-containing volatiles (Diallyl tetrasulfide, di-2-propenyl tetrasulfide; phenol, 2-(methylthio)-) being significantly correlated with glutathione metabolism (ko00480) enrichment. This reveals a feedback loop wherein ROS activates sulfur assimilation pathways to drive defensive sulfide production, establishing an “oxidative damage–sulfur volatiles release” regulatory axis.

### 5.2. N and K Co-Regulate Flavor Pathways

LN treatment significantly inhibits monoterpene synthase activity while activating glucosinolate metabolism, leading to enhanced pungent odors. This aligns with carbon flux redirection toward aromatic hydrocarbon degradation (ko00622) for carbon skeleton recycling under N limitation [57]—where accumulation of key intermediates (e.g., o-xylene) indirectly reduces phenylpropanoid/terpenoid precursor supply, suppressing floral–balsamic flavor formation. K deficiency activates flavonoid biosynthesis (ko00941) and toluene degradation (ko00623), promoting the upregulation of sweet compounds (e.g., 2-naphthyl methyl ketone) that synergize with metabolites possessing the woody characteristic (e.g., Spiro [4.5]dec-7-ene) to form a distinctive woody–sweet profile. Dual N–K deficiency co-suppresses shikimate–phenylpropanoid and glutathione metabolism (ko00480), diminishing ester/terpenoid synthesis capacity and further weakening floral–balsamic complexity. Carbon recycling represents a universal stress response. N deficiency uniquely activates sulfur metabolism, while K deficiency specifically enhances flavonoid pathways and sweetness. Under dual stress conditions, glutathione and ABC transporter functions are amplified, with a stronger inhibitory effect on esters and terpenoids. The persistent high expression of 2-naphthyl methyl ketone across the LN, LK, and LNLK treatment groups—functioning as a hub metabolite in phenylpropanoid networks [58]—drives sweet/floral enhancement by regulating cinnamate derivative synthesis. This cross-stress pattern indicates that plants optimize secondary metabolic resource allocation (e.g., sacrificing terpenoid synthesis to boost phenylpropanoid derivatives) to balance antioxidant defense and ecological signaling, constituting a universal adaptive strategy under nutrient limitation [59]. Future studies should integrate multiomics and molecular biology tools to enable the targeted manipulation of candidate genes, advancing *Panax* quality improvement from empirical practices to precision-driven designer breeding.

## 6. Conclusions

This study systematically elucidated the regulatory mechanisms of N, K, and LNLK on VOC composition and key aroma characteristics in ginseng roots through integrated GC–MS metabolomics, physio-biochemical analyses, and multivariate statistics. N/K deficiency significantly reduced root mineral elements (TN, TK, and TP) and OM content while inducing oxidative stress, as evidenced by O_2_^−^ and MDA accumulation along with elevated SOD/CAT activity. This physiological disruption triggered metabolic resource reallocation, resulting in significantly suppressed ginsenoside biosynthesis (reduced TGs and monomers such as Rb1/Rg1) while activating defense-related volatile pathways. These findings highlight a resource-optimizing “ginsenoside–VOCs trade-off” strategy. Analysis of the top 20 differential VOCs revealed that under LN conditions, terpenoids such as isoborneol (balsamic/camphor aroma) were significantly downregulated (*p* < 0.001), whereas sulfur-containing compounds such as Diallyl tetrasulfide, di-2-propenyl tetrasulfide (garlic/onion notes), were significantly upregulated (*p* < 0.01). This indicates enhanced pungency due to suppressed monoterpene synthesis and activated glucosinolate metabolism. Under LK, 2-naphthyl methyl ketone (sweet/orange blossom scent) was substantially elevated (*p* < 0.001) and served as a cross-stress marker persistently highly expressed in N-deficient and N–K-deficient groups, reflecting its role as a “metabolic hub” in phenylpropanoid networks that coordinates antioxidant defense, resource optimization, and ecological signaling—likely by regulating cinnamate derivative synthesis to enhance sweet-floral notes. Sulfur compounds (e.g., tetrasulfide and di-2-propenyl) upregulated in N-deficient treatments may improve stress adaptation through GSH-glucosinolate cycling. Thus, N/K nutrition dynamically shapes VOC profiles by differentially regulating phenylpropanoid, terpenoid, and sulfur metabolic pathways. Under LNLK, co-responsive metabolites exhibited functional integration: esters (e.g., acetic acid and phenylmethyl ester; sweet/floral) and terpenes (e.g., 2,6,6-trimethylcyclohexa-1,4-dienecarbaldehyde) were significantly downregulated (*p* < 0.05 and *p* < 0.001, respectively), suggesting synergistic suppression of the shikimate–phenylpropanoid pathway weakens floral–balsamic complexity. These findings demonstrate that secondary metabolite production constitutes a key adaptive mechanism for mitigating abiotic stress damage. This study establishes a theoretical framework for understanding how N and K interactions influence ginseng aroma quality, illustrating the metabolic adaptation of plants under nutrient stress as a survival advantage. Deciphering this mechanism not only highlights the wisdom of natural selection but also provides novel insights into precision fertilization strategies for ginseng cultivation. By optimizing fertilizer ratios to enhance desirable aromas (e.g., sweetness and florals) while suppressing off-odors, these findings offer practical applications for improving the commercial value of functional foods, regulating the quality of herbal medicines, and advancing research on the crosstalk between plant nutrition and secondary metabolism.

## Figures and Tables

**Figure 1 foods-14-02981-f001:**
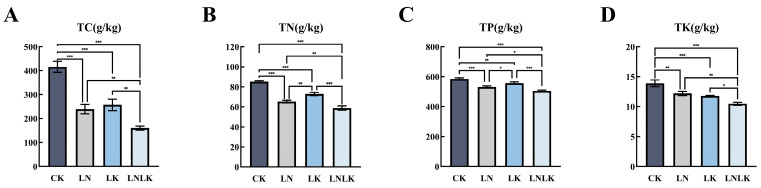
The effects of nitrogen and potassium deficiency on the nitrogen, phosphorus, potassium, and organic matter contents in the roots of *Panax ginseng*. Note: One asterisk (*) indicates a significance level of 0.05, two asterisks (**) indicate a significance level of 0.01, and three asterisks (***) indicate a significance level of 0.001. The same notation is applied hereinafter. (**A**) Organic matter content in ginseng roots. (**B**) Total nitrogen content in ginseng roots. (**C**) Total phosphorus content in ginseng roots. (**D**) Total potassium content in ginseng roots.

**Figure 2 foods-14-02981-f002:**
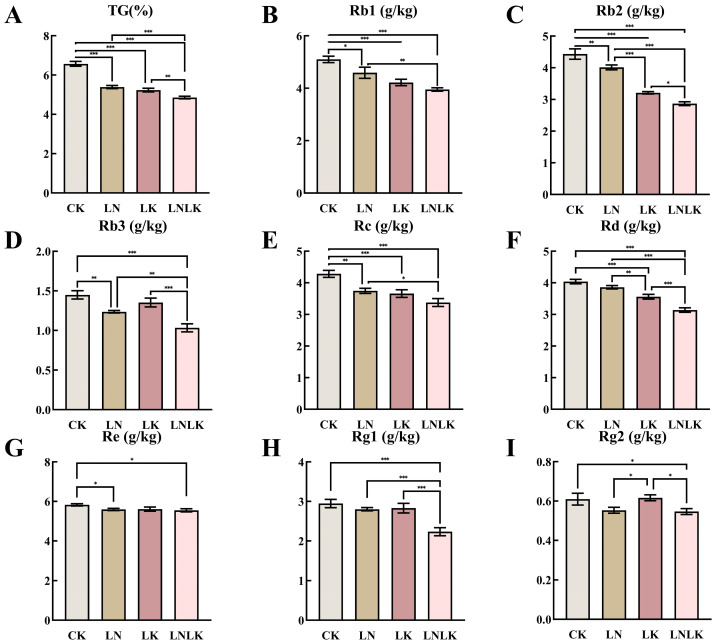
Effects of nitrogen and potassium deficiencies on ginsenoside content in the roots of *Panax ginseng*. (**A**) TS content in ginseng roots. (**B**) Rb1 content in ginseng roots. (**C**) Rb2 content in ginseng roots. (**D**) Rb3 content in ginseng roots. (**E**) Rc content in ginseng roots. (**F**) Rd content in ginseng roots. (**G**) Re content in ginseng roots. (**H**) Rg1 content in ginseng roots. (**I**) Rg2 content in ginseng roots. * *p* < 0.05, ** *p* < 0.01, *** *p* < 0.001.

**Figure 3 foods-14-02981-f003:**
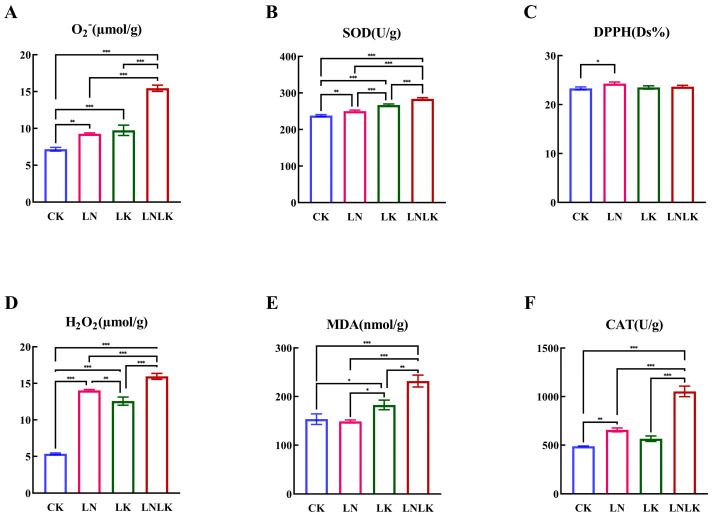
Effect of nitrogen and potassium deficiency on the antioxidant enzyme activity of ginseng. (**A**) O_2_^−^ content. (**B**) SOD content. (**C**) DPPH content. (**D**) H_2_O_2_ content. (**E**) MDA content. (**F**) CAT content. * *p* < 0.05, ** *p* < 0.01, *** *p* < 0.001.

**Figure 4 foods-14-02981-f004:**
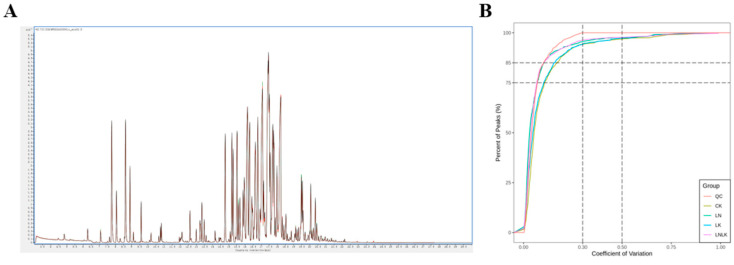
Quality control (QC) samples: mass spectrometry detection TIC overlap diagram and CV distribution diagram for each group of samples. (**A**) Total ion current (TIC) map of QC sample mass spectrometry analysis. (**B**) Coefficient of variation (CV) distribution map of metabolite peak areas for each group of samples. The horizontal axis represents CV values, and the vertical axis represents the proportion of substances with CV values less than the corresponding CV value out of the total number of substances. Different colors represent different groups of samples, with QC representing the quality control sample.

**Figure 5 foods-14-02981-f005:**
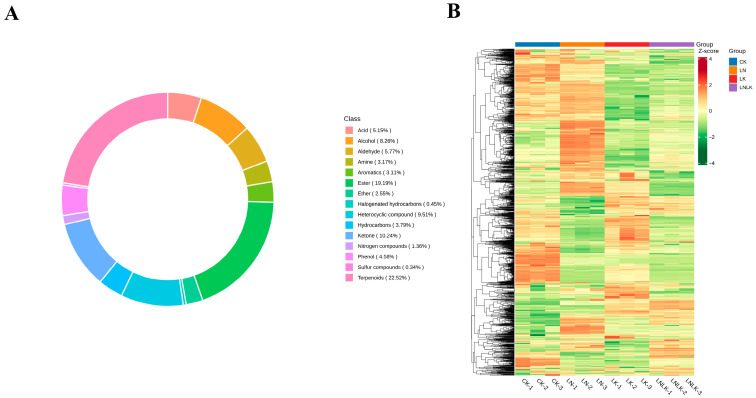
Circular plot of metabolite category composition and clustering heatmap of metabolite abundance. (**A**) Circular plot illustrating the composition of metabolite categories. (**B**) Hierarchical clustering heatmap of volatile metabolite abundance in ginseng roots under four treatments. The color gradient represents the accumulation levels of each metabolite, ranging from low (green) to high (red), with darker hues indicating higher metabolite content.

**Figure 6 foods-14-02981-f006:**
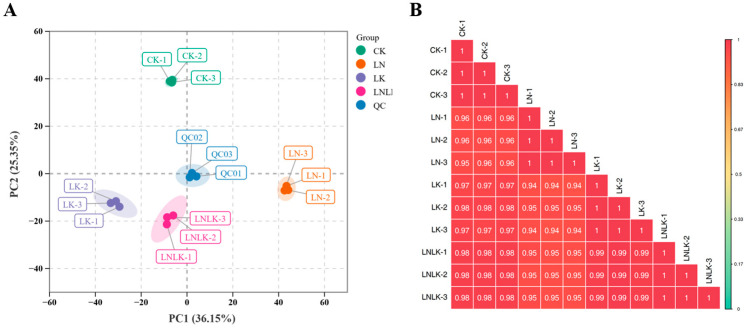
Plot of 2D results of PCA for all samples (including QC samples) and correlation between test samples. (**A**) Principal component analysis of the metabolites. (**B**) Correlation analysis between test samples.

**Figure 7 foods-14-02981-f007:**
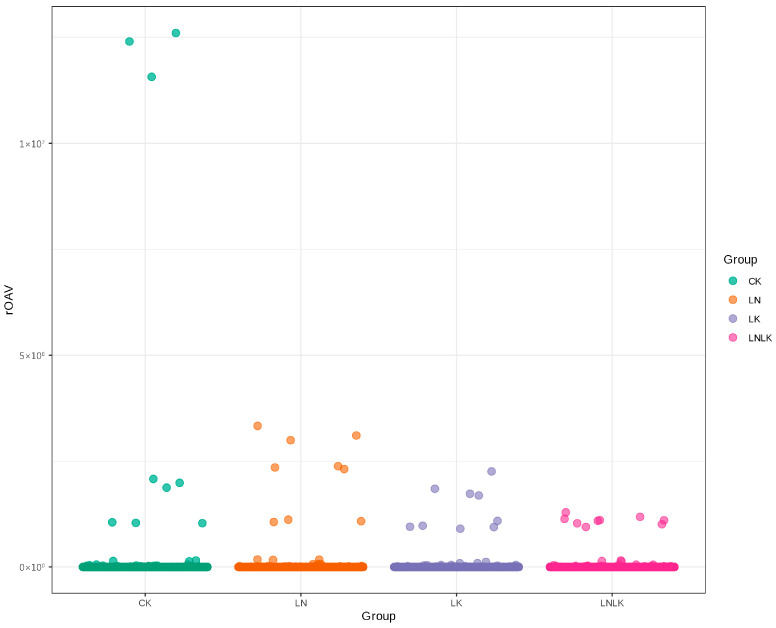
Scatter plot of rOAV odor activity values.

**Figure 8 foods-14-02981-f008:**
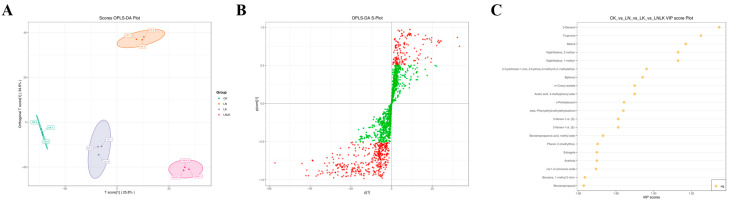
Corresponding group OPLS-DA score plot, corresponding group OPLS-DA S-plot, and VIP value plot of differential metabolites. (**A**) OPLS-DA score plot for the nitrogen-deficient group, potassium-deficient group, and nitrogen–potassium-deficient group. (**B**) OPLS-DA S-plot of N deficiency group, K deficiency group, and combined N–K deficiency group. (**C**) VIP value plot of differential metabolites in the N deficiency group, K deficiency group, and combined N–K deficiency group. Note: In the OPLS-DA S-plot, the horizontal axis represents the covariance between the principal component and the metabolite, and the vertical axis represents the correlation coefficient between the principal component and the metabolite. Metabolites closer to the upper right and lower left corners indicate more significant differences. Red dots indicate that the VIP value of these metabolites is greater than or equal to 1, while green dots indicate that the VIP value of these metabolites is less than 1. The VIP value plot of differential metabolites only shows the top 20 differential metabolites by VIP value.

**Figure 9 foods-14-02981-f009:**
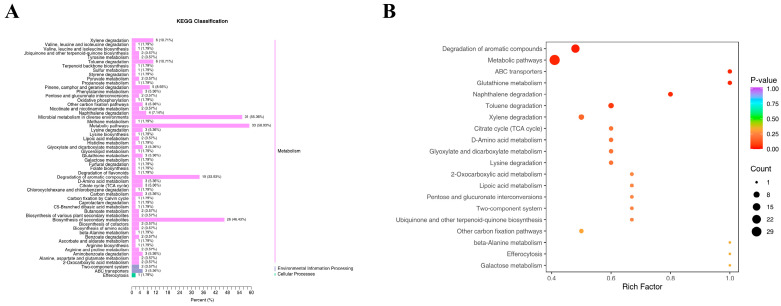
KEGG differential enrichment categorization map and KEGG differential enrichment bubble map. (**A**) KEGG differential enrichment classification plot. (**B**) KEGG differential enrichment bubble plot.

**Figure 10 foods-14-02981-f010:**
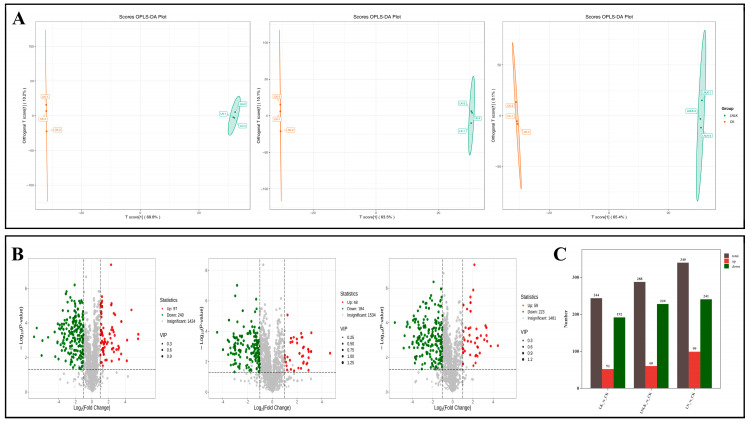
Corresponding subgroup OPLS-DA score plots, volcano plots, and statistics of the number of differential metabolites. (**A**) OPLS-DA score plots of volatile metabolites of ginseng roots among different treatment groups. (**B**) Volcano plot of differential metabolites among different treatment groups. (**C**) Statistical plots of the number of differential metabolites in ginseng roots under different treatments. Note: Each dot in the volcano plot represents a metabolite, and the size of the dot represents the variable importance (VIP) value in the project. In the volcano plot, the larger the absolute value of the horizontal coordinate, the more significant the differential expression is, and the more reliable the differential metabolites screened. The green dots in the plot represent downregulated differential metabolites, the red dots represent upregulated differential metabolites, and the gray dots represent the metabolites detected with no significant difference. In the graph of the number of differential metabolites, red represents the number of upregulated differential metabolites of the latter relative to the former in the comparison group, green represents the number of downregulated differential metabolites, and gray represents the total number of differential metabolites.

**Figure 11 foods-14-02981-f011:**
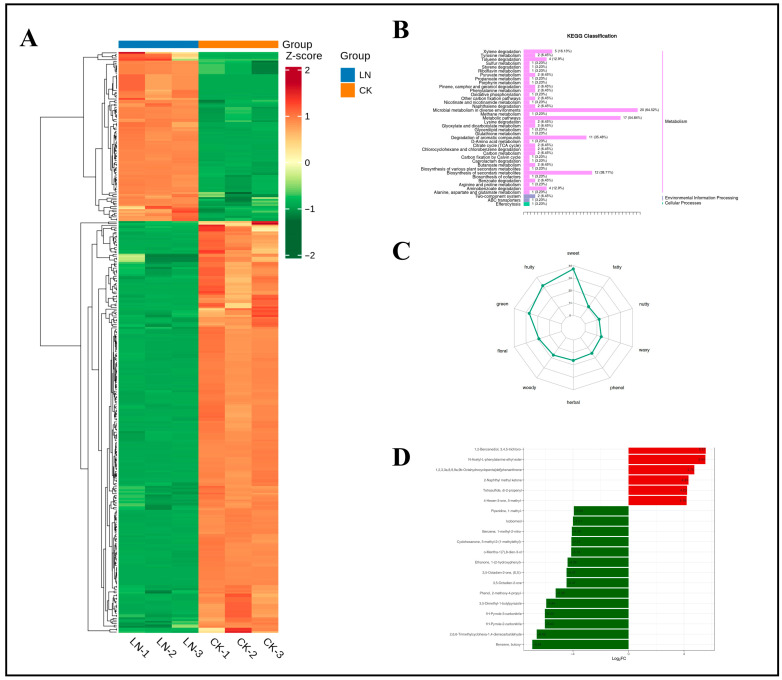
Clustering heatmap of differential metabolites under N deficiency, classification map of differential metabolite pathways, radar map of sensory flavor characterization of differential metabolites, and dynamic distribution of differences in the content of each of the top 20 substances corresponding to upward and downward grouping of differential changes. (**A**) Clustering heatmap of differential metabolites under nitrogen deficiency. (**B**) Classification map of differential metabolite pathways. (**C**) Radar plot of sensory flavor characterization of differential metabolites. (**D**) Bar chart of the top 20 substances with differential changes in differential metabolites. Note: The x-axis represents the log2FC of the differential metabolites; that is, the logarithm to the base 2 of the fold change of the differential metabolites. The y-axis represents the differential metabolites. Red indicates upregulation of metabolite content, and green indicates downregulation of metabolite content. In the KEGG diagram, the y-axis represents the names of metabolic pathways, and the x-axis represents the number of differential metabolites annotated to the pathway and the proportion of this number to the total number of annotated differential metabolites.

**Figure 12 foods-14-02981-f012:**
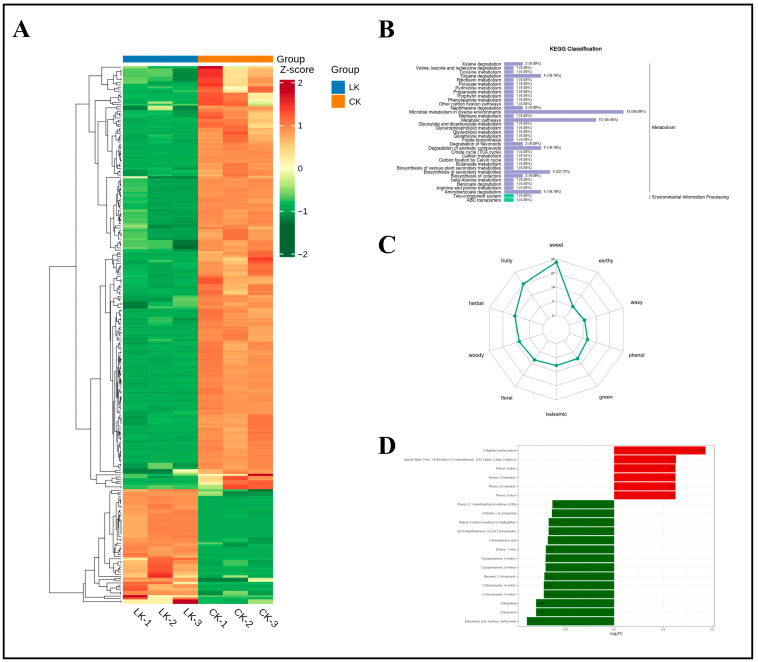
Clustering heatmap of differential metabolites under K deficiency, classification map of differential metabolite pathways, radar map of sensory flavor characterization of differential metabolites, and dynamic distribution map of differences in the content of each of the top 20 substances corresponding to upward and downward changes in group differences. (**A**) Clustering heatmap of differential metabolites under potassium deficiency. (**B**) Classification map of differential metabolite pathways. (**C**) Radar plot of sensory flavor characterization of differential metabolites. (**D**) Bar chart of the top 20 substances of differential changes in differential metabolites.

**Figure 13 foods-14-02981-f013:**
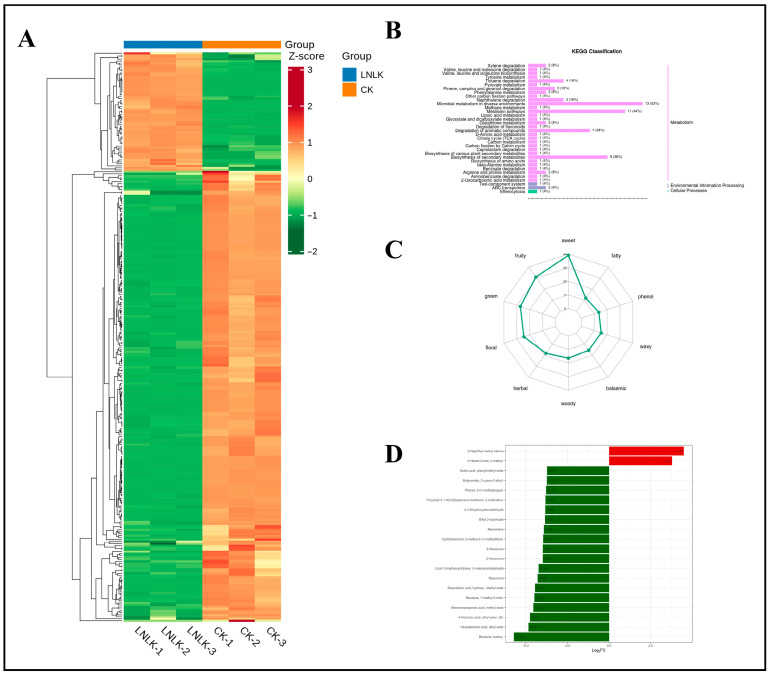
Clustering heatmap of differential metabolites under combined N–K deficiency, classification map of differential metabolite pathways, radar map of sensory flavor characterization of differential metabolites, and dynamic distribution map of differences in the content of each of the top 20 substances in the up and downregulation of differential changes. (**A**) Clustering heatmap of differential metabolites under combined N–K deficiency. (**B**) Classification map of differential metabolite pathways. (**C**) Radar plot of sensory flavor characterization of differential metabolites. (**D**) Bar chart of the top 20 substances with differential changes in differential metabolites.

**Table 1 foods-14-02981-t001:** Top 20 metabolites with the highest fold change under N deficiency.

Compounds	Class I	Aroma Description	LN	CK	Type
3,4,5-Trichloro-1,2-benzenediol	Alcohol	/	299,114 ± 10,978	6294 ± 675	Upregulated
N-Acetyl-L-phenylalanine ethyl ester	Ester	/	480,395 ± 23,484	10,291 ± 364	Upregulated
1,2,3,3a,8,9,9a,9b-Octahydrocyclopenta[def]phenanthrene	Aromatics	/	21,046 ± 229	785 ± 506	Upregulated
2-Naphthyl methyl ketone	Ketone	/	60,453 ± 39,320	3013	Upregulated
Diallyl tetrasulfide, di-2-propenyl tetrasulfide	Sulfur compounds	Strong garlic, onion	65,058 ± 3368	3478	Upregulated
5-Methyl-4-hexen-3-one	Ketone	/	4419 ± 913	242	Upregulated
1-Methylpiperidine	Heterocyclic compound	/	2064 ± 2043	31,688 ± 2792	Downregulated
Isoborneol	Terpenoids	Balsamic, camphor, herbal, woody	205,135 ± 9803	3296,923 ± 602,517	Downregulated
1-Methyl-2-nitrobenzene	Nitrogen compounds	/	1948 ± 69	33,174 ± 625	Downregulated
2-Isopropyl-5-methylcyclohexan	Terpenoids	Minty	1051 ± 88	18,311 ± 1088	Downregulated
2-Isopropenyl-3-methylenecyclohexanol	Terpenoids	/	1406 ± 71	24,753 ± 1538	Downregulated
1-(2-hydroxyphenyl)ethanone	Ketone	Phenol, sweet, hawthorn, tobacco, honey, herbal	12,547 ± 355	263,503 ± 41,314	Downregulated
3,5-Octadien-2-one, (E,E)-	Ketone	Fruity, green, grassy	1048	23,155 ± 701	Downregulated
3,5-Octadien-2-one	Ketone	Fruity, fatty, mushroom	1048	23,155 ± 701	Downregulated
2-methoxy-4-propyl-Phenol	Phenol	Clove, sharp, spicy, sweet, phenol, powdery, allspice	1403	53,602 ± 2355	Downregulated
1-Butyl-3,5-dimethyl-1H-pyrazole	Heterocyclic compound	/	3402	208,237 ± 11,490	Downregulated
1H-pyrrole-2-carbonitrile	Heterocyclic compound	/	1325	86,783 ± 13,060	Downregulated
1H-pyrrole-3-carbonitrile	Heterocyclic compound	/	1325	86,783 ± 13,060	Downregulated
2,6,6-Trimethylcyclohexa-1,4-dienecarbaldehyde	Aldehyde	/	856	84,078 ± 2016	Downregulated
Butoxybenzene	Ether	Anisic, licorice	1279	157,085 ± 6795	Downregulated

**Table 2 foods-14-02981-t002:** Top 20 metabolites with the highest fold change under K deficiency.

Compounds	Class I	Aroma Description	LK	CK	Type
2-Naphthyl methyl ketone	Ketone	Sweet, neroli, orange, blossom, neroli, powdery	76,556 ± 6746	3013	Upregulated
Spiro [4.5]dec-7-ene, 1,8-dimethyl-4-(1-methylethenyl)-, [1S-(1.alpha.,4.beta.,5.alpha.)]-	Terpenoids	/	11,157,201 ± 194,332	1254,135	Upregulated
Phenol, 3-ethyl- 3-Ethylphenol	Phenol	/	24,131 ± 1730	2754	Upregulated
Phenol, 2-ethyl- 2-Ethylphenol	Phenol	Phenol	24,131 ± 1730	2754	Upregulated
Phenol, 2,4-dimethyl- 2,4-Dimethylphenol;	Phenol	Weak, smoky, roasted, dark	24,131 ± 1730	2754	Upregulated
Phenol, 2,5-dimethyl- 2,5-Dimethylphenol;	Phenol	Sweet, naphthyl, phenol, smoky, bacon	24,131 ± 1730	2754	Upregulated
Phenol, (1,1-dimethylethyl)-4-methoxy- (CAS)	Phenol	Mild, rubbery	8860	78,641 ± 15,419	Downregulated
4-Penten-1-ol, propanoate	Ester	/	35,659	322,004 ± 27,654	Downregulated
Phenol, 4-chloro-5-methyl-2-(1-methylethyl)-	Phenol	/	334,552	3388,867 ± 275,723	Downregulated
2(1H)-naphthalenone, 3,4,5,6,7,8-hexahydro-	Ketone	/	109,080	1110,076 ± 228,5650	Downregulated
4-Aminobenzoic acid	Acid	/	1667	17,418 ± 971	Downregulated
Butane, 1-nitro- 1-Nitrobutane	Nitrogen compounds	/	3069	34,387 ± 1206	Downregulated
Cyclopentanone, 2-methyl- 2-Methylcyclopentanone	Ketone	Roasted, beefy	10,013	113,024 ± 772	Downregulated
Cyclopentanone, 3-methyl- 3-Methylcyclopentanone	Ketone	Roasted, beefy	10,013	113,024 ± 772	Downregulated
Benzene, (1-nitropropyl)- (1-Nitropropyl)benzene	Nitrogen compounds	/	8571	101,042 ± 9400	Downregulated
1,2-Benzenediol, 4-methyl- 4-Methyl-1,2-benzenediol	Alcohol	/	11,048 ± 10,959	133,103 ± 5893	Downregulated
1,2-Benzenediol, 3-methyl- 3-Methyl-1,2-benzenediol	Alcohol	/	11,048 ± 10,959	133,103 ± 5893	Downregulated
3-Hexanone	Ketone	Sweet, fruity, waxy, rummy, grape	571	9003 ± 594	Downregulated
2-Hexanone	Ketone	Fruity, fungal, meaty, buttery	571	9003 ± 594	Downregulated
Butanedioic acid, hydroxy-, diethyl ester Hydroxybutanedioic acid diethyl	Ester	Brown, sugar, sweet, wine, fruity, herbal	2359	51,612 ± 931	Downregulated

**Table 3 foods-14-02981-t003:** Top 20 metabolites with the highest fold change under combined N–K deficiency.

Compounds	Class I	Aroma Description	LNLK	CK	Type
2-Naphthyl methyl ketone	Ketone	Sweet, neroli, orange, blossom, neroli, powdery	66,718 ± 5049	3013	Upregulated
4-Hexen-3-one, 5-methyl- 5-Methyl-4-hexen-3-one	Ketone	/	3293 ± 250	242	Upregulated
Acetic acid, phenylmethyl ester	Ester	Sweet, floral, fruity, jasmine, fresh	16,299 ± 1803	215,710 ± 5917	Downregulated
Butyramide, 2-cyano-2-ethyl-	Amine	/	411	5472 ± 1295	Downregulated
Phenol, 2-(1-methylpropyl)-	Phenol	/	3703 ± 162	51,069 ± 845	Downregulated
Tricyclo [2.2.1.0(2,6)]heptane-3-methanol, 2,3-dimethyl-	Terpenoids	/	36,940 ± 686	521,286 ± 44,237	Downregulated
2,3-Dihydroxybenzaldehyde	Aldehyde	Almond, vanilla	27,327 ± 683	389,349 ± 15,554	Downregulated
Ethyl 2-octynoate	Ester	Violet, leafy, oily, waxy	3912 ± 168	56,235 ± 799	Downregulated
Memantine	Amine	/	90,870 ± 2709	1,372,610 ± 21,391	Downregulated
Cyclohexanone, 5-methyl-2-(1-methylethyl)- 2-Isopropyl-5-methylcyclohexan	Terpenoids	/	1177 ± 40	18,311 ± 1088	Downregulated
3-Hexanone	Ketone	Sweet, fruity, waxy, rummy, grape	571	9003 ± 594	Downregulated
2-Hexanone	Ketone	Fruity, fungal, meaty, buttery	571	9003 ± 594	Downregulated
2,6,6-Trimethylcyclohexa-1,4-dienecarbaldehyde	Aldehyde	/	4465 ± 226	84,078 ± 2016	Downregulated
Resorcinol	Phenol	Nutty, creamy, phenol, hawthorn, musty	1652	32,340 ± 2717	Downregulated
Butanedioic acid, hydroxy-, diethyl ester Hydroxybutanedioic acid diethy	Ester	Brown, sugar, sweet, wine, fruity, herbal	2359	51,612 ± 931	Downregulated
Benzene, 1-methyl-2-nitro- 1-Methyl-2-nitrobenzene	Nitrogen compounds	/	1482 ± 99	33,174 ± 625	Downregulated
Benzenepropanoic acid, methyl ester	Ester	Honey, fruity, wine, balsamic, floral	4482	105,109 ± 2242	Downregulated
4-Octenoic acid, ethyl ester, (Z)-	Ester	Fruity, fatty, green, pineapple, pear	1230 ± 51	33,135 ± 1314	Downregulated
Hexadecanoic acid, ethyl ester	Ester	Mild, waxy, fruity, creamy, milky, balsamic, greasy, oily	3857 ± 315	110,683 ± 10,046	Downregulated
Benzene, butoxy- Butoxybenzene	Ether	Anisic, licorice	2985 ± 2954	157,085 ± 6795	Downregulated

## Data Availability

Data will be made available on request.

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
