# Peer review of "Exploring the Effects of Nitrogen and Potassium on the Aromatic Characteristics of Ginseng Roots Using Non-Targeted Metabolomics Based on GC-MS and Multivariate Analysis"

_foods, 2025, doi:10.3390/foods14172981_

Round 1
Reviewer 1 Report
Comments and Suggestions for Authors
The article entitled “GC–MS Analysis of Nitrogen and Potassium Effects on Ginseng Root Aroma Profiles” presents a thorough investigation into how nitrogen (N) and potassium (K) deficiencies, both individually and in combination, affect the aroma profile of ginseng (Panax ginseng) roots, using metabolomic analysis based on gas chromatography–mass spectrometry (GC–MS).
Although the authors report the identification of a large number of volatile organic compounds (VOCs), specifically:
“In total, 1,768 metabolites were detected and classified into 15 categories (Figure 5A), including 398 terpenoids, 339 esters, 182 ketones, 168 heterocyclic compounds, 146 alcohols, 102 aldehydes, 91 carboxylic acids, 81 phenols, 67 hydrocarbons, 56 amines, 55 aromatic hydrocarbons, 45 ethers, 24 nitrogen-containing compounds, 8 halogenated hydrocarbons, and 6 sulfur-containing compounds”.
The section describing the materials and methods used for compound identification and quantification lacks sufficient detail.
More specifically:
- The authors state that compound identification was carried out using a custom-built database. However, no information is provided regarding how this database was constructed or validated. Details on its development and contents should be included.
- The quantification methodology is critically important, as it underpins the calculation of relative odor value activities (rOVAs). This section should be significantly expanded to clarify the quantification approach, including calibration procedures and any internal standards used.
- Furthermore, the authors should cite the sources of the odor threshold values used for the selected compounds. It is also necessary to specify the units used for Ci (concentration of the compound) and Ti (odor threshold).
Additionally, the quality of the figures should be improved. Several figures are too small to be legible, which hinders data interpretation and reduces the overall clarity of the manuscript.
Author Response
Thank you very much for taking the time to review this manuscript. Please find attached a detailed response, and the corresponding revisions have been marked in revision mode in the resubmitted file.

Reviewer 2 Report
Comments and Suggestions for Authors
The manuscript titled “GC–MS Analysis of Nitrogen and Potassium Effects on Ginseng Root Aroma Profiles” explores how deficiencies in nitrogen, potassium, or both affect the volatile metabolite composition and aroma profile of ginseng roots. Through the application of untargeted GC–MS-based metabolomics combined with multivariate statistics and rOAV analysis, the study provides novel insights into nutrient-induced shifts in volatile organic compounds (VOCs) and their sensory implications. This manuscript contains relevant and valuable findings. The introduction is well written, with a good literature search. However, revisions are needed. The names of VOCs must be corrected according to standard chemical nomenclature. Data tables require improved formatting, clearer labeling, and specification of measurement units. The manuscript should clarify how compounds were identified and quantified, including retention indices (RI) and software used. The quality of several figures is inadequate, making them difficult to interpret. Additionally, better visual formatting and consistency in terminology would enhance the manuscript’s readability and professionalism. Detailed comments are provided below.
Line 117: Are the authors certain that 0.05 mg of ginseng root powder was extracted? This value seems unusually low and should be double-checked.
Line 131: The procedure for determining total ginsenoside (TG) content should be described in more detail.
Lines 133–146: The paragraph on the determination of mineral elements is confusing. Initially, the authors describe digestion with HNO₃–HClOâ‚„, but later (lines 142–146) mention that this applies only to ICP-OES. How were samples prepared for the other methods: flame photometry, colorimetry, and elemental analysis?
Line 155-156: Replace “liquid N” with liquid nitrogen or liquid Nâ‚‚.
Line 157: Replace “20-mL headspace vial” with 20 mL headspace vial (no hyphen).
Lines 163–174: Please specify whether a solvent delay was used during GC–MS acquisition. Additionally, indicate the m/z range that was monitored.
Line 177: The authors state that compounds were identified by comparing mass spectra with a custom-built database. Was this identification performed automatically using specific software? Please clarify.
Line 219: In Figure 1, include a legend explaining the meaning of labels A, B, C, and D.
Line 233: As with Figure 1, a description of labels A through I should be added for clarity.
Line 270: The chromatogram image is of low resolution. Please replace it with a higher-quality version.
Line 271: The QC sample is introduced here for the first time. Please describe what it is and how it was prepared.
Line 280: The text in Figure 5 is not legible. Please improve figure resolution or layout.
Line 285: The authors report detection of 1,768 metabolites, which is an unusually high number of VOCs in a single sample. Most studies report only a few hundred compounds. The authors should explain this high count. Moreover, the Supplementary Material lacks information on retention times (RT) and retention indices (RI), which are essential for accurate compound identification, particularly for VOCs with similar mass spectra. Please also clarify what the numerical values in the Supplementary Tables represent. Are these total ion areas?
Line 516: Compound names appear to be generic names automatically generated from the GC–MS software database. These need to be revised to follow accepted chemical nomenclature. For example:
- “Tetrasulfide, di-2-propenyl” → Diallyl tetrasulfide, di-2-propenyl tetrasulfide
- “1,2-Benzenediol, 3,4,5-trichloro-” → 3,4,5-Trichloro-1,2-benzenediol
Also, compound classes in the tables are inconsistently labeled in singular and plural forms; please standardize.
There is a lack of units in the data tables. It is unclear whether the values represent relative percentage, area counts, or concentration (e.g., mg/g). Clarify this and include appropriate units. Some standard deviations are listed with two decimal places, which is unnecessarily precise. Round to one or no decimal places where appropriate. Additionally, several compounds show a standard deviation of zero, which is unrealistic and suggests either insufficient replicates or data processing issues. Note that the Materials and Methods section does not mention the number of replicates or whether measurements were repeated.
Line 519: Figure 12 is not readable due to low resolution; please replace or enhance it.
Line 577: The same issues apply to Table 1 (unclear units, format).
Line 579: Figure 13 is also of poor quality; improve the resolution.
Line 638: As with Table 1, Table 3 suffers from formatting issues. Also, there are typographical errors (e.g., broken words like terpenoi ds). Tables should span the full-page width to improve legibility and prevent text from breaking into multiple lines. Additionally, the column titled “Type” is unclear; please rephrase using “Upregulated” or “Downregulated.” As previously noted in the text, the short forms should be added in brackets for more clarity.
Line 752: The names of Supplementary Materials (e.g., Figure S1, Table S1) should be listed with proper titles. Refer to the journal’s Guide for Authors for formatting guidelines.
Author Response

(The authors gave the same response as above.)

Round 2
Reviewer 1 Report
Comments and Suggestions for Authors
The authors have answered the questions posed. The article can be published in its current form.
Author Response
Thank you!
Reviewer 2 Report
Comments and Suggestions for Authors
The authors applied required changes in the manuscript. I suggest further processing this article for publication.
The only dilemma I have is about adding RT for detected compounds. Most of the articles published in scientific journals include RT for detected compounds. Additionally, because the authors worked in SIM mode, they should include the m/z transitions that they monitored. But I assume that they would consider this information confidential as well.
Since this is an open access journal, I think this information would be important for readers, but I leave the decision of whether it is necessary for the article to the Editor.
Author Response
Thank you!